# Effect of Cooking and Domestic Storage on the Antioxidant Activity of Lenticchia di Castelluccio di Norcia, an Italian PGI Lentil Landrace

**DOI:** 10.3390/ijerph20032585

**Published:** 2023-01-31

**Authors:** Mattia Acito, Cristina Fatigoni, Milena Villarini, Massimo Moretti

**Affiliations:** Unit of Public Health, Department of Pharmaceutical Sciences, University of Perugia, Via del Giochetto, 06122 Perugia, Italy

**Keywords:** lentil, Castelluccio di Norcia, antioxidant activity, cooking procedures, storage, plant-based diets

## Abstract

The aim of this work was to characterise Lenticchia di Castelluccio di Norcia (an Italian PGI lentil landrace) and assess the impact of cooking and storage on antioxidant activity. After opening the package (T_0_), samples were analysed using a set of chemical assays (i.e., total phenolic content, DPPH, ABTS, and ORAC assays). Analyses were also conducted on boiled, pressure-cooked, and 6-month-stored (T_1_) products. At both T_0_ and T_1_, raw Lenticchia di Castelluccio di Norcia PGI showed higher total phenolic content (T_0_: 9.08 mg GAE/g, T_1_: 7.76 mg GAE/g) and antioxidant activity (DPPH T_0_: 33.02 µmol TE/g, T_1_: 29.23 µmol TE/g; ABTS T_0_: 32.12 µmol CE/g, T_1_: 31.77 µmol CE/g; ORAC T_0_: 3.58 μmol TE/g, T_1_: 3.60 μmol TE/g) than boiled and pressure-cooked samples. Overall, pressure-cooking led to a smaller decline in total phenolic content and antioxidant activity than the common boiling procedure. Domestic storage led to a significant reduction in total phenolic content—both in raw and cooked products—but not in antioxidant activity. In summary, these results highlighted interesting amounts of phenols and antioxidant properties of this product, showing the impact of routine procedures. Given the relevance of pulses as sustainable plant-based meat alternatives and their importance in the prevention of non-communicable diseases, health professionals should consider these aspects in the context of correct nutrition education and scientific communication.

## 1. Introduction

Lenticchia di Castelluccio di Norcia (LCN) (*Lens culinaris* Medik.) is a Protected Geographical Indication (PGI) lentil landrace that is farmed at 1400 m above sea level in the Apennine Mountains on the border between the Marche and Umbria Regions (Central Italy) [1]. This landrace is completely adapted to the local pedoclimatic characteristics of this area and, therefore, can be perfectly managed by adopting low-input agriculture practices. It is characterised by highly heterogeneous and very small seeds (2 mm in diameter). Ten subpopulations differing in seed coat colour (light and dark green, brown, pink, and grey), type of pattern (dotted or marbled), and cotyledon colour (yellow or orange) have been identified within the LCN ecotype; this is a unique peculiarity among Italian landraces [2,3]. Moreover, specific molecular analyses showed that LCN can be unequivocally identified using genomic simple sequence repeat (SSR) markers and is still distinct from other lentil materials after 20 years of PGI certification [4].

Along with other pulses, lentils represent a sustainable plant-based source of protein and a good alternative to meat consumption, which is currently better accepted by consumers when compared with other alternative protein sources, such as insects and cultured meat [5]. Moreover, lentils represent an important source of complex carbohydrates, fibres, minerals, vitamins, polyphenols, and other phytochemicals with several potential health benefits, including anticarcinogenic potentialities, blood pressure-lowering and hypocholesterolemic, and glycemic load-lowering activity [6]. However, these features can vary considerably among different varieties [7,8,9,10]. Therefore, given the large number of cultivars and landraces, general observations about lentil characteristics might sound somewhat limited.

Despite its local popularity as an ingredient of traditional soups, little is known about the potential nutraceutical activity of LCN. Very little data are available in the literature, concerning some quality traits of LCN lentil seeds, including protein content (22–26.8%), ash (2.9%), mineral composition (P, 4.0–4.3 mg/g; K, 9.2–9.4 mg/g; Ca, 0.6–1.2 mg/g; Mg, 1.3 mg/g; Na, 0.24 mg/g; Fe, 0.07–0.08 mg/g; Cu, 1.6 mg/g), bioactive compounds (ascorbic acid, 4.5 mg/100 g; vitamin E, 28 µg/g; carotenoids, 16 µg/g; total phenols, 124 mg/100 g; total flavonoids, 103 mg/100 g), and antioxidant activity (ABTS assay, ≈40% inhibition; DPPH assay, ≈5% inhibition; ferrozine-based assay, ≈10% inhibition of the ferrozine-Fe^2+^ complex; FRAP assay, ≈250 µM TE/g) [3,11]. However, these studies uniquely investigated raw products. As lentils are exclusively consumed as cooked by humans, it would be crucial to also carry out extensive characterisations on cooked products. Moreover, there is the actual possibility that, in domestic conditions, a part of packaged lentils is stored for several months—once the package has been opened—before being consumed, even respecting the best-before date. This is a common issue in everyday life that might represent a source of variability in the products’ features [12,13,14].

In this context, the main aims of this work were as follows:(i)Characterisation of LCN in terms of total phenolic content (TPC) and antioxidant activity;(ii)Assessment of the effect of both boiling and pressure-cooking procedures;(iii)Assessment of the effect of domestic storage on the product’s characteristics.

## 2. Materials and Methods

### 2.1. Chemicals, Reagents, and Media

All reagents used were of analytical grade. Methanol and sodium carbonate (Na_2_CO_3_) were purchased from Carlo Erba Reagenti Srl (Milan, Italy). 2,2-Diphenyl-1-picrylhydrazyl (DPPH), 6-hydroxy-2,5,7,8-tetramethyl chromane-2-carboxylic acid (Trolox), Folin–Ciocâlteu reagent, hydrochloric acid (HCl), and gallic acid (GA) were obtained from Sigma-Aldrich Srl (Milan, Italy). The ABTS Antioxidant Capacity Assay Kit was obtained from Bioquochem S.L. (Llanera, Spain). The OxiSelect™ ORAC Activity Assay Kit (Catalog Number, STA-345) was purchased from Cell Biolabs, Inc. (San Diego, CA, USA). Distilled water was used throughout the experiments.

### 2.2. Plant Material

Lenticchia di Castelluccio di Norcia (LCN) PGI was grown in Castelluccio di Norcia (42°49′31.3″ N 13°12′32.0″ E, Municipality of Norcia, Umbria Region, Italy) in 2018, following the traditional low-input agricultural technique. Lentils were sowed in April and harvested in August. After harvesting, seeds were dried, packaged in 250 g sealed cardboard/plastic boxes (Appendix A), and provided for analysis. Seeds were identified by Agricultural Genetics researchers at the University of Perugia (Department of Agricultural, Food and Environmental Sciences, Unit of Agricultural Genetics and Genetic Biotechnologies).

### 2.3. Extraction Method, Cooking Procedures, and Domestic Storage

To extract phenolic compounds, lentil samples were extracted according to the method described by Moretti and co-workers, with minor modifications [15]. Briefly, seeds (1 g) were ground into a flour powder (24,000 rpm, 60 s), using IKA^®^ Tube Mill Control (IKA^®^-Werke GmbH & Co. KG, Staufen, Germany), and placed in a conical flask and extracted at room temperature for ca. 13 h with 20 mL of 70% methanol containing 0.1% HCl (*v*/*v*), using a magnetic stirrer (Stuart Stir US151, Cole-Parmer, Stone, UK). The mixture was centrifuged (3075× *g*, 10 min), and the supernatant was collected and stored at +4 °C, while the residue was re-extracted twice more using the same procedure. Finally, the supernatants were put together and dried using a vacuum rotary evaporator (STRIKE 300, Steroglass S.r.l., Perugia, Italy) and reconstituted in the same solvent used for extraction (20 mL). Extracts were stored at −20 °C in glass tubes until use.

In order to assess the effect of boiling and pressure-cooking on total phenolic content and antioxidant activity, seeds (100 g) were boiled in tap water (2 L) for 30 min in a common pot or for 15 min in a pressure-cooker (La Classica Lagofusion^®^, Lagostina S.p.A., Omegna, Italy) (55 kPa, 112 °C). Boiling and pressure-cooking time (half the time chosen for the boiling procedure) were indicated by local producers. Thirty minutes for common boiling was also indicated in the literature [3]. The cooked mass corresponding to 1 g of dry seeds was then homogenised (24,000 rpm, 20 s), freeze-dried (using Christ^®^ ALPHA I/5, Martin Christ Gefriertrocknungsanlagen GmbH, Osterode am Harz, Germany) for 24 h and extracted using the same procedures adopted for the raw product. Cooking water (50 mL) of LCN at T_0_ was also collected, centrifuged (3075× *g*, 10 min), and directly analysed.

The extraction procedure was also carried out 6 months (T_1_) after the package opening (T_0_) in order to assess the effect of domestic food storage on the product’s characteristics. Domestic storage conditions were reproduced by opening the product case, then later fixing it with adhesive tape and keeping it in a dark cupboard at room temperature for 6 months.

### 2.4. Total Phenolic Content

Total phenolic content (TPC) was assessed using the Folin–Ciocâlteu method, and quantification was based on a calibration curve generated using gallic acid (GA) (10–500 µg/mL) [15]. Briefly, 25 µL of GA standard or lentil extract was mixed with 125 µL 0.2 M Folin–Ciocâlteu reagent in a 96-well plate for 10 min at room temperature. Then, 125 µL of 2% (*m*/*v*) sodium carbonate (Na_2_CO_3_) solution was added, and the resulting mixture was incubated for 30 min. The absorbance was read at 765 nm using an Infinite^®^ 200 PRO microplate reader (Tecan Italia Srl, Milan, Italy), and the results were expressed as mg of GA equivalents (GAE) per gram of dry matter (DM) (mg GAE/g DM).

### 2.5. DPPH Assay

The assessment of the radical scavenging capacity of the lentil extracts was carried out according to a previously described method [15]. Briefly, 25 µL of lentil extract or the Trolox standard solution (125–2000 µM) was mixed with 200 µL of a methanolic solution of DPPH (350 µM) in a 96-well plate. After 30 min (at room temperature), the absorbance was read at 517 nm using a Sunrise microplate reader (Tecan Italia Srl, Milan, Italy). The DPPH antioxidant activity was reported as µmol Trolox equivalent (TE) per gram DM of lentil (µmol TE/g DM).

### 2.6. ABTS Assay

The 2,2′-azino-bis(3-ethylbenzothiazoline-6-sulfonic acid) radical cation assay was performed by using the ABTS Antioxidant Capacity Assay Kit (Bioquochem S.L., Llanera, Spain), following the manufacturer’s instructions. Briefly, 5 µL of antioxidant standard (ascorbic acid) or lentil extract was mixed with 200 µL of the ABTS•+ solution (Abs 0.70) in a 96-well microplate and reacted for 5 min at room temperature while continuously stirring. The absorbance of the reaction mixture was read at 734 nm using an Infinite^®^ 200 PRO microplate reader (Tecan Italia Srl, Milan, Italy). Calibration was achieved with an aqueous ascorbic acid solution (0–600 µM). The inhibition (%) of absorbance was calculated as:Inhibition (%) = [1 − (A_f_/A_0_)] × 100
where A_0_ is the absorbance of the control (uninhibited radical cation) and A_f_ is the absorbance of the sample after 5 min of incubation. The antioxidant activity was reported as µmol vitamin C Equivalent (CE) per gram of dry matter (µmol CE/g DM), based on the calibration curve (inhibition of absorbance of standards plotted as a function of their concentrations).

### 2.7. ORAC Assay

The oxygen radical absorbance capacity (ORAC) assay was carried out according to the manufacturer’s instructions on a 96-well plate using the OxiSelect™ ORAC Activity Assay Kit (Cell Biolabs, Inc. San Diego, CA, USA). Fluorescein was used as the fluorescent probe, and fluorescence was recorded every 5 min for a total of 60 min (excitation wavelength, 480 nm; emission wavelength, 520 nm) at 37 °C using an Infinite^®^ 200 PRO microplate reader (Tecan Italia Srl, Milan, Italy). The results were calculated by plotting the net area under the curve (AUC) against the Trolox (standard) concentration. The results were expressed as µmol Trolox equivalent (TE) per gram DM of lentil (µmol TE/g DM), based on the standard curve calibration (0–50 µM).

### 2.8. Statistical Analysis

Tests were performed on lentil seeds coming from two different packs (replicates) provided by lentil producers. For each replicate, TPC and DPPH assays were carried out in triplicate (replications in 96-well plate), whereas ABTS and ORAC assays were carried out in duplicate (replications in 96-well plate). The results were expressed as mean values ± standard error of the mean (SEM). Cooking waters were analysed on a qualitative level. For the other tests, data were analysed on a significance level of 0.05 with one-way ANOVA, and a post hoc analysis was carried out with the Dunnet and Bonferroni tests. The correlation analysis between TPC, DPPH, ABTS, and ORAC was expressed by calculating the coefficient of determination (R^2^) and taking into consideration the values obtained in the raw, boiled, and pressure-cooked product. All calculations were carried out using SPSS (IBM Corp. Released 2010. IBM SPSS Statistics for Windows, Version 19.0. Armonk, NY, USA: IBM Corp.) statistical software for Windows. GraphPad Prism software (GraphPad Software, Inc., San Diego, CA, USA) was used as the graphics programme.

## 3. Results

### 3.1. Total Phenolic Content

The total phenolic content of LCNs is shown in Figure 1 and Appendix A. At T_0_, the average TPC in raw Umbrian lentils was 9.08 mg GAE/g DM. Both cooking procedures led to a statistically significant loss in phenolic compounds (*p* < 0.05).

In every sample analysed, we found that boiling and pressure-cooking procedures led to a significant decrease in TPC. At T_0_, boiled LCN showed TPC values of 4.94 mg GAE/g DM; whereas, when pressure-cooked, its value was 5.29 mg GAE/g DM.

Our results showed that pressure-cooking led to a smaller decline in TPC than common boiling procedures. As proof, compared with the boiling procedure, cooking water obtained after pressure-cooking showed lower TPC values, confirming the hypothesis of an inferior loss in phenolic compounds when adopting this cooking technique in this species. In particular, the cooking water TPC value after the boiling process was 0.23 mg GAE/mL (corresponding to 4.52 mg GAE/g DM), whereas the cooking water TPC value after pressure-cooking was 0.15 mg GAE/mL (3.03 mg GAE/g DM).

After 6 months of domestic storage, in raw and cooked LNC, TPC values were significantly lower than those observed at T_0_. However, an appreciable amount of phenolic compounds was still found, even after storage and cooking procedures.

### 3.2. DPPH Assay

The antioxidant activity was first assessed with the DPPH assay (Figure 2 and Appendix A). The DPPH value of raw LNC (T_0_) was 33.02 µmol TE/g DM. This study found significantly higher DPPH values in pressure-cooked samples than in boiled samples (*p* < 0.05).

The cooking water showed remarkable antioxidant activity, with DPPH values of 1.02 µmol TE/mL (20.45 µmol TE/g DM) and 1.00 µmol TE/mL (19.94 µmol TE/g DM) in boiled and pressure-cooked samples, respectively.

When comparing T_0_ and T_1_, the results showed that antioxidant activity was significantly decreased by storage only in the raw sample.

### 3.3. ABTS Assay

Similar trends to those observed with the DPPH assay were confirmed in almost all of the tests performed using the ABTS scavenging capacity assay (Figure 3 and Appendix A). Specifically, cooking procedures (boiled: 8.14 µmol CE/g DM; pressure-cooked: 10.28 µmol CE/g DM) led to a reduction to approximately one-fourth and one-third, respectively, of the ABTS values observed in the raw sample (32.12 µmol CE/g DM). At T_0_, the ABTS values of pressure-cooked samples were significantly higher than those reported for boiled samples. Moreover, 6 months of domestic storage did not affect ABTS values in either raw or cooked samples, but in this case, values between boiled and pressure-cooked samples were not significantly different.

As observed in the DPPH assay, antioxidant activity was detected in both boiled (1.17 µmol CE/mL, corresponding to 23.48 µmol CE/g DM) and pressure-cooked (1.10 µmol CE/mL, corresponding to 21.93 µmol CE/g DM) LCN cooking waters, with slightly lower values found in pressure-cooked samples.

### 3.4. ORAC Assay

ORAC values of lentil samples are shown in Appendix A. The antioxidant activity measured by the ORAC assay showed trends that were comparable to those of the other tests performed. Cooking procedures led to an overall reduction in antioxidant activity. As seen in the ABTS assay, domestic storage did not affect the antioxidant activity measured by this test.

Finally, even in this case, both cooking water samples showed detectable antioxidant activity (boiled: 0.02 µmol TE/mL, corresponding to 0.39 µmol TE/g DM; pressure-cooked: 0.02 µmol TE/mL, corresponding to 0.36 µmol TE/g DM).

### 3.5. Correlation Analysis

The correlation between the analytical methods used was also determined by taking into consideration the values obtained in the raw, boiled, and pressure-cooked product (Table 1). At both T_0_ and T_1_, DPPH, ABTS, and ORAC assays demonstrated a high linear correlation with TPC, suggesting that phenolic compounds strongly contributed to the antioxidant activity of this product.

## 4. Discussion

In this study, we carried out a characterisation of LCN in terms of TPC and antioxidant activity, focusing also on the effect of both boiling/pressure-cooking procedures and on the effect of domestic storage on the product’s characteristics.

The raw LCN TPC values we have found are higher than those reported in several Canadian lentils by Tsao and co-workers [16], who performed analogous extraction procedures and used the same solvent employed in this study. The TPC values are also higher than those reported for LCN in a study investigating the impact of soil salinity on lentil characteristics [11]. However, the extraction procedures and solvents used were not the same as those used in our study; therefore, differences in results should be considered with caution. In every sample, we found that boiling and pressure-cooking procedures led to a significant decrease in TPC. Our findings are in line with those obtained by Xu and Chang, who observed a significant loss in TPC values of lentils after both boiling and pressure-cooking [17]. Analogously, another work showed a significant decline in TPC after boiling procedures in seven commercialised lentils [18]. Conversely, our data are in contrast with Di Maro and co-workers, who analysed methanolic extracts of a southern Italy lentil landrace and reported a negligible loss of TPC in cooking water [19]. However, as the choice of different extracting solvents has been shown to dramatically affect the yields of phenolic compounds in pulses [20], a comparison with other reports using different extraction procedures is not always appropriate.

We found that pressure-cooking led to a smaller decline in TPC than common boiling procedures. As proof, compared with the pressure-cooking procedure, cooking water obtained after common boiling showed higher TPC values, confirming the hypothesis of a higher leach of phenolic compounds when adopting this cooking technique for this species. The leaching of phenolic compounds in cooking water after the boiling procedure amounted to approximately 50%, which is similar to trends observed in other species belonging to the Fabaceae family [15,21].

Similar to TPC, antioxidant activity was deeply affected by cooking procedures, which was in line with another work which analysed several lentil varieties [18]. Moreover, we analogously found significantly higher DPPH values in pressure-cooked samples than in boiled samples. These findings are in line with another study [22] which showed that the boiling procedure reduced antioxidant activity (assessed using DPPH and ABTS assays) more than the pressure-cooking procedure. Several works highlighted that, as a consequence of the thermal processing of pulses, the content of phytochemicals with antioxidant activity usually decreases [23]. This should be taken into consideration in the context of nutritional counselling and health communication. Pulses are indeed recognised among the most interesting and sustainable alternatives to meat protein and are included in the most renowned international diet guidelines for the prevention of non-communicable diseases [24,25]. However, as pulses are usually consumed after a cooking treatment, information on the loss of bioactive compounds, as well as the choice of the best cooking technique, should also be included within these recommendations. It should be noted that—as observed in our study as well—processed pulses often retain remarkable amounts of such compounds, maintaining their potential health benefits after being cooked [23].

Similar trends to those observed with the DPPH assay were confirmed in almost all of the tests performed using the ABTS scavenging capacity assay. Even in this case, cooking waters showed remarkable antioxidant activity. As suggested by other authors [21], it can be hypothesised that antioxidant compounds released in cooking water were not destroyed/inactivated by thermal processing or, alternatively, were broken into smaller compounds, still showing antioxidant activity. These findings described lentil cooking water not only as a mere by-product but also as a potentially underestimated source of bioactive compounds and may open new scenarios for further nondietary, health-related employment.

Overall, LCN was found to be not so sensitive to 6-month domestic storage. Although we observed a slight (but significant) decrease in TPC, these compounds were also found in stored samples in non-negligible amounts, which probably might lead, along with other non-phenolic antioxidant compounds, to the global conservation of antioxidant properties.

Our study also has some limitations. A qualitative and quantitative analysis (e.g., LC-MS/MS, etc.) of single polyphenols and other antioxidant compounds has not been carried out. However, based on the results we have obtained, this investigation deserves to be addressed in future research.

Finally, the correlation analysis between TPC, DPPH, ABTS, and ORAC was expressed by calculating the coefficient of determination (R^2^). The antioxidant assays we used demonstrated a high linear correlation with TPC, suggesting that phenolic compounds strongly contributed to the antioxidant activity of this product. This is in line with previous works, showing that TPC results correlate well with several antioxidant assays used in food analysis, including DPPH, ABTS, and ORAC [26].

## 5. Conclusions

In summary, these results highlighted interesting chemical features of this product, showing the impact of routine procedures. With only a few exceptions, pressure-cooking generally led to a smaller decline in total phenolic content and antioxidant activity than the common boiling procedure. Domestic storage led to a significant reduction in total phenolic content—both in raw and cooked products—but not in antioxidant activity.

Given that lentils represent one of the most appreciated plant-based meat alternatives and have great potential in the prevention of non-communicable diseases, these aspects should be taken into consideration by health professionals in the context of nutrition education and health communication.

## Figures and Tables

**Figure 1 ijerph-20-02585-f001:**
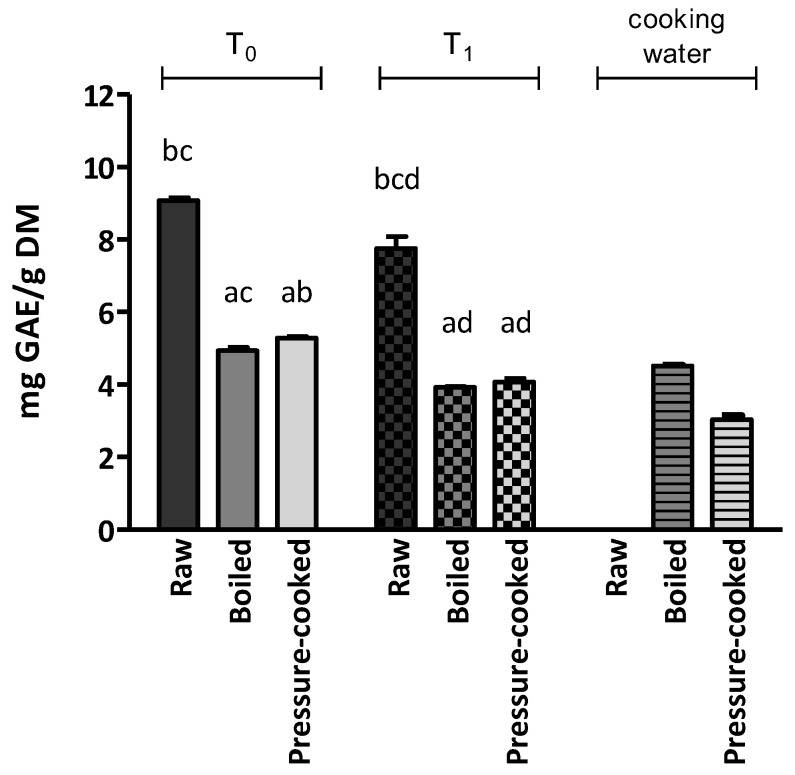
Total phenolic content of Lenticchia di Castelluccio di Norcia PGI at T_0_, T_1_ and its cooking water. The results are expressed as the mean ± standard error of the mean. Statistical significance: *p* < 0.05. ^a^ vs. raw; ^b^ vs. boiled; ^c^ vs. pressure-cooked; ^d^ vs. T_0_.

**Figure 2 ijerph-20-02585-f002:**
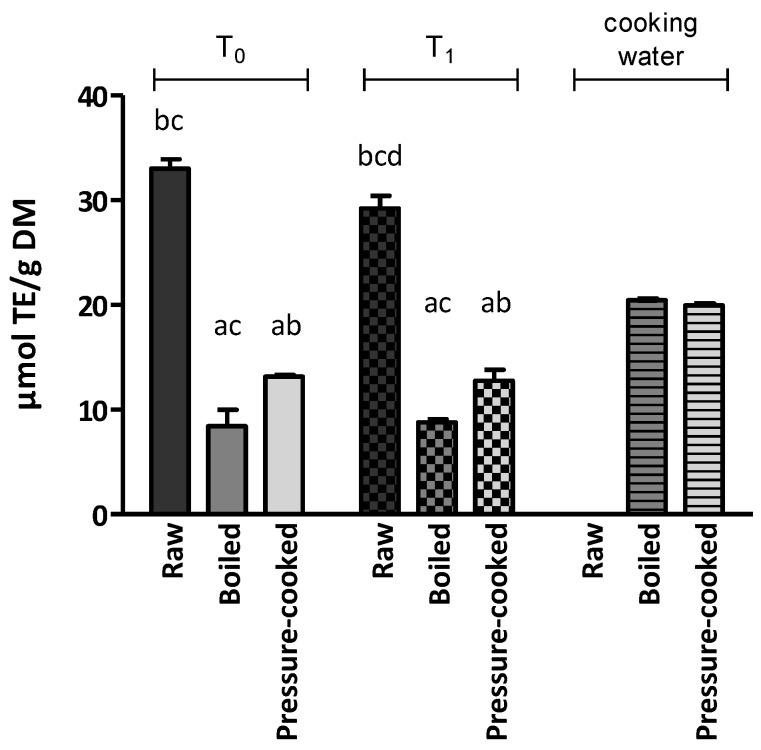
Antioxidant activity of Lenticchia di Castelluccio di Norcia PGI at T_0_, T_1_ and its cooking water assessed by the DPPH assay. The results are expressed as the mean ± standard error of the mean. Statistical significance: *p* < 0.05. ^a^ vs. raw; ^b^ vs. boiled; ^c^ vs. pressure-cooked; ^d^ vs. T_0_.

**Figure 3 ijerph-20-02585-f003:**
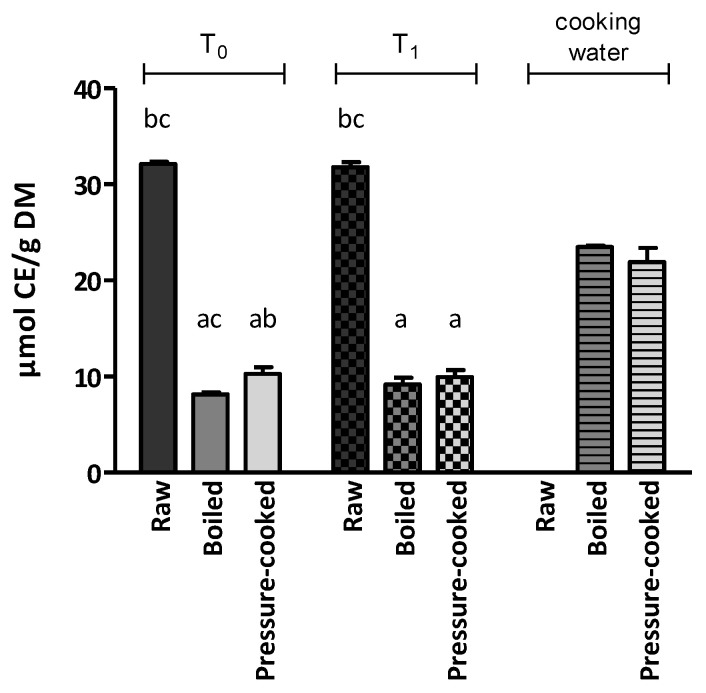
Antioxidant activity of Lenticchia di Castelluccio di Norcia PGI at T_0_, T_1_ and its cooking water assessed by ABTS assay. The results are expressed as the mean ± standard error of the mean. Statistical significance: *p* < 0.05. ^a^ vs. raw; ^b^ vs. boiled; ^c^ vs. pressure-cooked.

**Table 1 ijerph-20-02585-t001:** Correlation coefficients (R^2^) between total phenolic content (TPC), DPPH, ABTS, and ORAC assays at T_0_ and T_1_.

**T_0_**	**TPC**	**DPPH**	**ABTS**	**ORAC**
**TPC**		0.9890	1.0000	0.9998
**DPPH**	0.9890		0.9897	0.9917
**ABTS**	1.0000	0.9897		0.9999
**ORAC**	0.9998	0.9917	0.9999	
**T_1_**	**TPC**	**DPPH**	**ABTS**	**ORAC**
**TPC**		0.9778	1.0000	0.9345
**DPPH**	0.9778		0.9765	0.8425
**ABTS**	1.0000	0.9765		0.9367
**ORAC**	0.9345	0.8425	0.9367	

## Data Availability

The data presented in this study are available within the article.

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
