# Peer review of "Effect of Cooking and Domestic Storage on the Antioxidant Activity of Lenticchia di Castelluccio di Norcia, an Italian PGI Lentil Landrace"

_ijerph, 2023, doi:10.3390/ijerph20032585_

Round 1
Reviewer 1 Report
This manuscript is not new for the study. The treatment was done with the known knowledge. The correlation of the total phenolic content and antioxidants are highly related in general. The details of significant contents of introduction till conclusion should be shown more and concisely. English written style should be improved in academic format.
Author Response
Please, see attached file

Reviewer 2 Report
Thank you for the opportunity to review the work entitled “Effect of cooking and domestic storage on the antioxidant activity of Lenticchia di Castelluccio di Norcia, an Italian PGI lentil landrace”. I have read this work with interest, and I don’t have major comments here. However, I would like to ask a few questions about the used methods and the number of samples that were investigated, i.e:
Dear Authors,
1. Please, remove the repetition in the line 72: “Distilled water was used throughout the 71 experiments.”
2. In general, one information is missing for the methods, although although there are some indications in the statistics section. How many packages were opened (replicates)? How many samples were boiled (replicates)? How looks the n number for each stage of the methodology for extracts and antioxidant properties?
3. You have specified that "the assays were carried out in sextuplicate or in quadruplicate". Does it mean the number of replications (opened packages, prepared/boiled samples or extracts) or the number of replications per 96-well plate?
4. Why have you not checked polyphenols and antioxidant traces in cooking water after T1?
I have no further comments regarding the introduction section as well as the description of the results and discussion. The conclusions are also correct. Despite the above-mentioned comments about the methods, the procedures themselves are very legible (including the ranges of the used standard curves) and make it easy to replicate the protocols without consulting other sources. If the OxiSelect™ ORAC Activity Assay 147 Kit has any SKU number, then please provide it in the “2.6. ORAC Assay” section.
I wish you a successful work on corrections.
Author Response
Please, see attached file

Reviewer 3 Report
Accordin my personal opinion discussion should e made on laboratory results. these may be more or less in agreement with the bibliography, they must be the result of a scientific study where each question is answered with measured results. Here it sounds to me that some data relating to the experimentation carried out are missing.
Data collected is sufficient to characterize the Lentil of Castelluccio di Norcia if it is not compared to other commercial cultivars in terms of TPC content.
Result discussion shoulb be complete, but according my opinion there is a lack of experimental information that cannot be compensated for by bibliographic comparison.

Author Response
Please, see attached file

Reviewer 4 Report
This manuscript evaluated the effects of two processing methods( boiled and pressure cooked) on total phenol and antioxidant activities of Lenticchia di Castelluccio di Norcia. Meanwhile, the antioxidant activity before and after 6 months of storage was compared. This study provided guidelines for consuming lentils in Italy. However, only total phenols was observed, which seems to be insufficient to support the conclusion. The discussion is relatively simple.
Here are some issues below:
1. Line 19-20 Please describe specific chemical features.
2. Keyword: Please delete the keywords and “human health”, which are not covered in detail in this article.
3. Please add description on chemical composition in lentils and their health implications in Introduction.
4. Please specify the packing of the lentils. Will they be packed vacuum or sealed? The packaging material?
5. delete “Distilled water was used throughout the experiments. ” in line 72.
6. Line 95- 96 Please explain why the pressure cooking time is set at half of the cooking time? Also specify the conditions of pressure cooking, such as handling pressure.
7. Line 102- 103 Please explain the reason for choosing a storage period of 6 months and specify whether the storage period starts from the opening of the package or after the plants have been harvested.
8. Line 104-106, about the “Domestic storage conditions”, the case was opened once throughout the next 6 months? It seems different from that of real life situation.
9. Line 172 Please note the significant difference (P < 0.05 or 0.01)
10. It’s recommended combine Fig.1 a, b, and c into one column chart. Also, the labeling of statistical results is complicated and confused. Pls verify the figures throughout the whole manuscript.
11. Please explain why there is no difference between the two cooking methods after 60 days of storage?
12. Line 245-247 Please explain the reason why only correlation analysis between indexes under different storage times was analyzed instead of correlation analysis under boiling and pressure cooking.
13. Line 256 Please rewrite “The experimental results highlighted interesting features of this landrace.” this sentence. Please specific analysis the interesting features of this landrace
14. Please analyze the reasons for the higher TPC and antioxidant activity in water in part of Discussion. Please add the extraction method of cooking water to the test method.
15. Please explain why the index of total phenol was selected for the study, instead of other components with antioxidant activity in lentils. It’s better to describe the nutritional components of lentils in introduction.
16. In the discussion, it seems that the author's analysis of the effect of storage time on the antioxidant activity of Italian PGI lentil landrace was not found. Please add this part of the analysis and explain the reason why the TPC content of lentils decreased but the antioxidant activity did not change significantly after 6 months of storage.
17. Line 310-312 This is not consistent with the results of the previous analysis of antioxidant activity on ABTS.
18. Line 314-317 was repeated with Line 21-23. Please pay attention to the difference between the conclusion and the abstract. Rewrite this sentence.
Author Response
Please, see attached file

Round 2
Reviewer 3 Report
The subject dealt with deserves further study in scientific and analytical terms.The study of the effect of cooking and conservation could be relevant when treated on a large and varied number of samples. The analytical approach is meager, recourse is made to methods that the scientific bibliography has amply demonstrated to be obsolete and insignificant. I believe that the scientific content is too little to be able to define this laboratory exercise as a scientific work
Author Response
Please, see the attached file.

Reviewer 4 Report
The authors has answered my questions and provided more detailed information to improved the quality of the manuscript. I agree to publish it in the journal.
Author Response
Please, see the attached file.
